# Systemic Modeling of the Peace–Development Nexus

Bernard Amadei 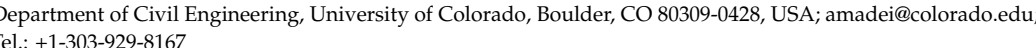

Department of Civil Engineering, University of Colorado, Boulder, CO 80309-0428, USA; amadei@colorado.edu;
Tel.: +1-303-929-8167

**Abstract:** As we enter the third decade of the 21st century, the value proposition of promoting sustainability and peace in the world has become more imperative than ever. It is an appropriate time to pause and reflect on what a post-pandemic COVID-19 world will look like and what constitutes a new mindset toward a more sustainable, stable, peaceful, and equitable world where all humans live with dignity and at peace. As emphasized in this paper, the new mindset must acknowledge that sustainability and peace are two entangled states of dynamic equilibrium. It is hard to envision a sustainable world that is not peaceful and a peaceful world that has not endorsed sustainable practices. This paper looks more specifically at the value proposition of adopting a systems approach to capture the linkages between selected development sectors (e.g., SDGs) and peace sectors (e.g., positive, negative, and cultural). Basic system dynamics (SD) models are presented to illustrate the peace–development nexus dynamics. The models are general enough to be used for different contexts and scales.

**Keywords:** peace–development nexus; SDGs; system dynamics; mindset; COVID-19 pandemic

## 1. Introduction

The year 2020 is likely to be remembered as a pivotal moment in socio-economic development worldwide. Starting with an outbreak believed to have started at the end of 2019, COVID-19 has evolved into a "democratic super disease" [1] that "respects no borders" [2]. The pandemic has revealed existing forms of vulnerability, injustice, and inequality; pushed more people into poverty [3]; and impacted peace negatively worldwide [4].

The pandemic has shown the fragility, interconnectedness, and diversity of human life and the close linkages between humans and their environment. It has negatively affected all the systems involved in socio-economic development and negatively disrupted public life, predominately for humanity's most vulnerable sections. As summarized in the 2020 sustainable development report [5], "the world is facing the worst public health and economic crisis in a century". The World Bank predicts a "lost decade" for the world economy [6]. On the flip side, the pandemic "has brought out some of the best human characteristics; self-sacrifice in helping others; empathy and solidarity despite the need for social distancing" [7]. It is also a "wake-up call and a training ground to enhance our joint and resilient response to future pandemics and other external disturbances" [7].

The pandemic has also stressed the challenges in planning, designing, and implementing humanitarian and development interventions and programs. According to the Alliance for Peace Building [8], "In 2020, nearly 168 million people needed humanitarian assistance and protection—about 1-in-45 people in the world—the highest figure in decades". Combined with the many planetary challenges already existing before 2020 (e.g., population growth and migration, urbanization, climate change, and environmental protection, among many others), the pandemic has added confusion and uncertainty about intervening appropriately in the systems (social, economic, cultural, ecological, and technical) deemed responsible for these challenges. There is a realization that the uncertainty and predictability of these systems can only be handled using systems thinking and tools of complexity science. In the context of a post-2020 world, systems thinking can be used to (i)

capture various socio-economic development dynamics, (ii) explore different intervention scenarios, and (iii) develop integrated and collaborative solutions that transcend local and national boundaries [9].

As we enter the third decade of the 21st century and navigate through what Tooley [10] referred to as "VUCA (volatile, uncertain, complex and ambiguous) times", it is appropriate to pause and reflect on (i) what a post-pandemic world would look like for a growing and more urbanized world's population [11]; (ii) how to build back better (rather than return to the old normal) more resilient and equitable inclusive societies that put all people back at the center of human and economic development [12]; and (iii) how to build capacity and increase resilience at different scales (household, community, national, and regional) to cope and adapt to future adverse events and challenging conditions (e.g., climate change and natural disasters). Together, answering these questions will hopefully contribute to creating a much anticipated sustainable and peaceful world for all.

Since 1990, many authors and organizations have advocated the need for a paradigm shift in the way humans interact with each other and the environment on which they depend. The overarching goal of building a more sustainable, stable, peaceful, and equitable world where all humans live with dignity and peace is not new. It has been in the mind of many constituencies during the 20th century, starting with the League of Nations (1920–1946), followed by the United Nations ever since. It was the underlying thrust behind the publication of *Common Future* [13] and *Agenda 21* [14], and in establishing the Millennium Development Goals (1990–2015) and, more recently, the Sustainable Development Goals (SDGs, 2015–2030) agenda, also known as Agenda 2030. This agenda represents a comprehensive plan of action that involves five critical aspects of sustainability: people, planet, prosperity, partnerships, and peace [15].

The noble vision of a more sustainable, peaceful, equitable, and stable world emphasized by the SDGs since 2015 is worthy of consideration. Paraphrasing Albert Einstein, "Problems cannot be solved with the same level of thinking [mindset] that created them"; a question arises as to whether the original SDGs' vision and its associated mindset and precepts are still relevant after 2020 or need to be updated. There seems to be a consensus on the need to readjust socio-economic development to reflect the new normal of a post-2020 world [16,17]. As noted by Sachs et al. [5], "COVID-19 will have several negative impacts on most SDGs". Implementing the SDGs agenda and mitigating these impacts may take more years to implement than planned in 2015 or even early 2020.

As noted by UNDP [3], the value proposition of sustainability and peace in the world of tomorrow has become more imperative than ever. Simply put, it is hard to envision a peaceful and sustainable world in the foreseeable future with yesterday's normal(s). It is time to acknowledge that "a world divided by wealth and poverty, health and sickness, food and hunger, cannot long remain a stable [and peaceful] place for civilization to thrive" [18]. More than ever, science, technology, and innovation (STI) have a critical role to play in creating a peaceful and sustainable world for all [7].

This paper presents first some insights into the new development mindset for a post-2020 world. It builds on the joint dynamic of peace and development discussed in the article by Amadei [19]. The paper looks more specifically at using system dynamics to model the linkages and dynamic interactions between some development and peace sectors of interest rather than modeling how to achieve a specific or a combination of several SDGs. The generic models presented herein can explore the peace–development nexus in a high abstraction and strategic level of decision-making and implementation and different contexts and scales. However, the models require policymakers and practitioners to develop a systems thinking mindset first and be willing to explore the various interdependencies between peace and development.

## 2. A New Post-2020 Mindset

A new mindset is urgently needed to implement humanitarian and development interventions and programs in a post-2020 world. This recommendation is based on Mead-

ows' [20] observation that changing the mindset in any system represents the place (i.e., leverage) to intervene with the highest return on the investment. However, such intervention is not as easy as it sounds, since the new mindset components must first be clearly outlined, understood, and adopted by multiple stakeholders (e.g., insiders and outsiders in development projects and programs) before they are implemented and assessed over time. In general, adopting a new mindset is initially challenging since it implies behavior change, which takes time and can be difficult for certain groups of stakeholders, policymakers, and practitioners involved in humanitarian and development aid.

It is interesting to note that the recommendation for a new mindset is already part of the post-2020 vernacular literature discussion. It is common to read, for instance, that changing the mindset for many constituencies is about "adopting a new normal" and "pivoting". This naïve concept ignores the fact that there was never a one-size-fits-all normal to start with before the pandemic but rather multiple normal(s), some better than others. In addition, pivoting is fine if there is a vision attached to it. Without it, there is a danger of ending up pointing in the original direction. With that in mind, creating a more sustainable, stable, and equitable world where all humans live with dignity and peace must be done in an intelligent, systemic, fair, and compassionate manner where normal is seen "as a plural". This approach departs from the dominant neoliberal capitalistic one that was the de facto mindset of the 20th century and the first two decades of the 21st century, characterized by determinism, compartmentalization, fear, greed, and benefit a few [21].

Let us explore some of the many characteristics of the new mindset that humanity must embrace to address the challenges mentioned above and adequately handle potential future crises (natural or human-made). First, the mindset must acknowledge that the challenges facing humanity cannot be tackled in isolation. They are complex and involve multiple interconnected components (social, economic, cultural, and technical) specific to the context and scale of the landscape in which they unfold. At times, these components may even transcend national boundaries requiring regional and international collaboration. The uncertainty, ambiguity, and unpredictable nature of the systems involved in the various challenges facing humanity imply that a systems/integrated approach to sustainability and peace is better suited in the new mindset than a deterministic one to capture their dynamics and linkages, explore different intervention scenarios, and develop integrated solutions. There is enough evidence that repeating the past business-as-usual mindset of ignoring the complexity of the systems at play in socio-economic development and treating them in isolation has the potential to do more harm than good [22].

A second aspect of the new mindset is that it requires humanity to reconsider its values and socio-economic priorities (i.e., how it sees reality) and put them into practice. A more mature level of consciousness in the day-to-day management and operation of our institutions and our occupations [23–25] is needed. As noted by Tanabe [2], "a critical question is emerging that faces humanity as a whole: what should come in the first place—society or economy, strong public health or profit, citizen's physiological, psychological, intellectual and spiritual well-being or plutocracy". Another remark is how to incentivize "each human citizen as a critical and transformative agent to contribute building sustainable global peace". These two recommendations represent a departure from the traditional human development dynamics where citizens are passive actors subject to policies that "maximize the economic advantage [for a strong capitalist class] while directing little energy to humanity's social, cultural, and even spiritual self-improvement or maturity" [2].

Third, the COVID-19 pandemic has demonstrated "the power of [scientific research] collaboration to create solutions quickly" [26]. The new mindset must build on such success stories and emphasize the importance of innovation in human and economic development at different scales (individual, household, community, country, regional, and planetary). As suggested by TWI2050 [7], "new thoughts, frameworks, and methods for the STI [Science, Technology, and Innovation] ecosystem to promote innovation, efficiency, and sufficiency for the achievement of the SDGs" are needed. Innovation must lead to solutions that embody the five aspects of Agenda 2030, i.e., they must be good for people

and the environment, be profitable, promote human security and social justice, and create meaningful and just partnerships [27].

Fourth, collective activities at different scales from local to international are urgently needed to prevent further decline in human development in a post-COVID-19 world [3,5]. According to Moritz [28], to avoid further instability worldwide, socio-economic partnerships and collaboration must simultaneously address short-term improvements to the current situation and long-term sustainability planning along five tracks: (1) repair what is currently most damaged; (2) rethink change without going back to how things were, i.e., without rebuilding the vulnerability of business-as-usual; (3) reconfigure change so that it can happen; (4) restart change with the recommendations mentioned above; (5) and report how change progresses with the ability for course correction through monitoring and evaluation. Simply put, yesterday's socio-economic development tools have a limited range of applications in developing the world of tomorrow. The metaphor of not placing new wine into old wineskins but using new wineskins [29] comes to mind. Innovative development tools and priorities are needed to operate in a new socio-economic structure.

An open-ended question arises as to how Moritz's five changing tracks affect how the SDGs and their respective targets and indicators must be addressed now and in the future. Another aspect of these five tracks is that they emphasize the importance of capacity building and resilience at different scales (individual, household, community, country, region, and global) in the overall discussion on sustainable development for the 21st century. These two concepts are not new and have been part of the development vocabulary for a long time. A traditional approach to building capacity and resilience is to identify and address in a fragmented manner specific issues at play in the systems (institutional, economic, social, environmental, and infrastructure) that may prevent the delivery of services and meeting particular SDGs. This compartmentalized approach, driven by a need to reach some form of satisfactory equilibrium, fails to account for possible states of synergy and trade-offs at play between these systems and the changing, adaptive, and dynamic nature of social networks [30].

Since the 2020 pandemic is a "wake-up call and a training ground to enhance our joint and resilient response to future pandemics and other external disturbances" [7], the concepts of capacity and resilience must be reimagined, redefined, and strengthened to handle future crises (e.g., health). As discussed further in a paper by Amadei [31], a systemic approach to capacity and resilience is needed to explore possible synergies and trade-offs at play in humanitarian and development interventions in a specific context and scale. At the community scale, capacity and resilience cut horizontally across multiple vertical silos of community development. One type of capacity building to achieve a specific goal, such as providing a reliable service (e.g., water, energy, food, transportation, etc.), could affect achieving another goal associated with a different service type over time. Likewise, the interaction between various systems at play (e.g., institutional, sociocultural, infrastructure, environmental, economic, etc.) at some scale may contribute to resilience when exposed to small and large adverse events over time.

Fifth, the dynamic between insiders and outsiders involved in development and humanitarian projects and programs needs to be participatory. Until about 30 years ago, the Western world's traditional approach to humanitarian and development work was top-down contractual and consultative, with limited input from the bottom-up beneficiaries [32,33]. More recently, there has been more emphasis on promoting collaborative and collegial approaches and transformation through empowerment [32,33]. Yet, development and humanitarian work shaped by external actors remains a dominant way of doing things today.

Sixth, the new mindset must recognize the linkages between sustainability and peace. Simply put, it is hard to envision a sustainable world that is not peaceful and a peaceful world that has not endorsed sustainable practices. Peace is fully integrated into the sustainable development agenda with SDG 16 (Peace, Justice, and Strong Institutions), which is to "Promote peaceful and inclusive societies for sustainable development, provide access to justice for all and build effective, accountable and inclusive institutions at all levels" [5].

Meeting SDG 16 is about the right relationships, which is contained in principle 16f of the Earth Charter [34], defining peace as " ... the wholeness created by right relationships with oneself, other persons, other cultures, other life, Earth, and the larger whole of which all are a part". According to Gittins and Velasquez-Castellanos [35], such relationships' core characteristics include freedom and "empathy, acceptance, and honesty". The National Peace Academy [36] sees the right relationships as part of a peace system that encompasses five interactive spheres of peace at the personal, social, political, institutional, and ecological levels.

Finally, the new mindset must also include the inner dimension of human development, i.e., what Maslow [37] referred to as esteem and self-actualization. It is interesting to note that such needs are not explicitly emphasized in the SDGs agenda, which prioritizes addressing outwardly the bottom tiers of Maslow's pyramid (i.e., meeting physiological and safety needs). This inner dimension is critical if the SDGs need to be met and the world does not revert to past practices. A quote of Meister Eckhart [38] to support the inner dimension of human development is pertinent here: "the outer work can never be minor when the inner work is a major one, and the outer work can never be major or good when the inner work is a minor one and without value". With that in mind, one can question the quality of our western society's inner work, institutions, and decision-makers based on how such groups (including ourselves) have managed planetary challenges over the past 200 years. Lessons can be learned from less well-known non-Western forms of human development that emphasize strengthening the whole person's inner dimension concurrently with socio-economic development.

## 3. An Integrated Approach to the SDGs

The 17 SDGs and associated targets and indicators were introduced in 2015 as a new 15-year long road map for worldwide sustainable development at the country level. In launching the so-called Agenda 2030, the General Assembly of the United Nations "recognize[d] that eradicating poverty in all its forms and dimensions, including extreme poverty, is the greatest global challenge and an indispensable requirement for sustainable development" [15]. Compared to the preceding Millennium Development Goals (1990–2015), the SDGs apply to all countries regardless of their development level.

Fulfilling the SDGs by 2030 is indeed a daunting task, a truly formidable undertaking, unparalleled in human history. Since 2015, the comprehensive SDGs agenda has been a work in progress, and the SDGs' targets and indicators have been refined further. Significant additions include introducing the SDG index and dashboards to quantify the progress of different countries on the SDGs [39] and introducing six SDG transformations to operationalize the SDGs' implementation at the country level [7,40]. They include (i) education, gender, and inequality; (ii) health, well-being, and demography; (iii) energy decarbonization and sustainable industry; (iv) sustainable food, land, water, oceans; (v) sustainable cities and communities; and (vi) digital revolution for sustainable development. The 2020 sustainable report [5] proposes short-term and long-term guidelines to address these six transformations considering the COVID-19 pandemic. Despite such efforts, the jury is still out on how the SDGs' targets and indicators need to be updated individually and together to match the post-COVID-19 world's new reality.

Another question that has always been pertinent to the SDGs agenda since its inception is how to meet the goals across different physical scales (country, cities, communities, households, and individuals) and temporal scales (short-, medium- and long-term). Scharlemann et al. [41] remarked that progress toward sustainable development might have synergistic benefits at one physical or temporal scale but create negative impacts, requiring trade-offs, at other scales.

Since 2015, there has been an increasing interest in understanding and quantifying how the SDGs interact with each other, since sustainable development is more than meeting a series of independent goals [42]. Zelinka and Amadei [43] noted that addressing the connections among the SDGs in a multi-sectoral integrated approach is crucial to ensure

the coherence of Agenda 2030. Although the SDGs represent an "indivisible whole" [44], some goals are likely to affect others positively (i.e., creating synergies) or negatively (i.e., requiring trade-offs). In contrast, others may only have indirect interactions or no interaction at all. Furthermore, as noted by Scharlemann et al. [41], the nature of the linkages across the SDGs being considered depends on (i) the context and "groups of actors" at the country level; and (ii) the perspective (socio-economic, geopolitical, geographic) used to explore the interactions (e.g., the environment-human linkage perspective).

The socio-economic development literature is rich in contributions that emphasize, mostly qualitatively, the value proposition of using an integrated approach to Agenda 2030. Landmark papers, among many others, include those of Griggs et al. [44]; Nilsson et al. [45]; Griggs et al. [46]; Waage et al. [47]; Coopman et al. [48]; Vladimirova and Le Blanc [49]; Barbier and Burgess [50]; Morton et al. [51]; Lim et al. [52]; and TWI2050 [53]. Qualitative and semi-quantitative tools have been used to quantify the SDGs' interactions [54–58]. A noteworthy contribution to understanding the SDGs' interdependence is the report entitled *A Guide to SDGs' Interactions* [59]. The report examined the interdependence at the target level between SDGs # 2 (zero hunger), 3 (good health and well-being), 7 (energy), and 14 (life below water) with the other goals, using a semi-quantitative impact factor ranging over a seven-point scale: neutral impact (0); different levels of positive impact (+1 to +3); and different levels of adverse impact ($-1$ to $-3$). More recently, Scharlemann et al. [41] provided an extensive review of the different formulations proposed in the literature since 2015 that capture the interaction between the SDGs, emphasizing other interaction types between humans and their environment. Many of these formulations use a double-causality analysis, which, in the case of the 17 SDGs, consists of creating a $17 \times 17$ matrix describing the direct influence and dependence of each SDG on the other goals. The analysis would become more complicated if one were to analyze the SDGs' interaction at the target level.

Quantitative tools borrowed from systems science have also been proposed to model the SDGs' interactions. They include, for instance, neural network analysis [60,61], cross-impact analysis [43], and system dynamics [62]. As summarized by Zelinka and Amadei [43], using such tools to address the SDGs has a strong value proposition when exploring how complexity and uncertainty in the country's systems affect the decision-making process of policymakers and practitioners. More specifically, using the habits of systems thinking developed by the Waters Foundation as a guide [63], systems science tools can be used to:

- Make meaningful connections across the overall Agenda 2030 and within and between the various systems that affect the SDGs.
- Understand the big picture of sustainable development at the country level while simultaneously paying attention to specific details.
- Explore and evaluate different perspectives in the eyes of various stakeholders involved in decision-making from the private and public sectors, civil societies, and others.
- Appreciate how different mental models of development shape views and actions when addressing the SDGs and selecting strategies.
- Recognize that sustainable development is a dynamic process that requires flexible and adaptive decision-making while recognizing patterns and trends.
- Explore the role that assumptions in decision-making play in shaping outcomes.
- Realize that the structure of the systems involved in sustainable development influences its dynamic.
- Account for time-delays (short and long-term) between making country-level decisions and observing the associated outcomes and how such delays require monitoring and evaluation.
- Consider possible intended and unintended implications of decision-making and policies.
- Explore the role of some structural variables, archetypes, reinforcing and balancing feedback loops of cause and effect, and patterns of sustainable development play in shaping emerging behavior and identifying possible leverage points in meeting the SDGs with a higher return on actions taken.
- Identify how accumulations and rates of change control the behavior of multiple systems.

- Realize that sustainable development requires a flexible and adaptive approach to decision-making, leading to good enough solutions.
- Recognize that multiple strategies, approximations, parametric and sensitivity studies need to be considered before coherent solutions are outlined, evaluated, and an implementation plan can be selected and implemented.
- Better understand how meeting the SDGs (and their targets) depends on the initial country capacity and resilience and the potential for capacity building over time.
- Predict how countries may respond to different strategies of capacity development under constraints and disturbances.

In summary, using a systems approach to address the interactions among the SDGs (and their targets) represents an alternative to the traditional deterministic and rigid decision process used in development worldwide over the past 50 years. As discussed in the previous section, a prerequisite for using such an approach is that decision-makers and practitioners involved in development interventions and programs must be willing to adopt a new mindset of systems thinking when working in partnership with other stakeholders. Unfortunately, the history of human and economic development over the past 50 years shows that a lack of will often hinders any change and progress [7].

## 4. The Peace–Development Nexus

Since its inception, Agenda 2030 has acknowledged the coherence between peace and development and that "there can be no sustainable development without peace and no peace without sustainable development" [15]. The two are entangled. Understanding and modeling the peace–development nexus requires some preliminary discussion about peace. This section summarizes key concepts about peace necessary to understand the narratives behind the system dynamics simulations presented below.

The peace studies and conflict management literature is rich in contributions exploring the different aspects of peace. Peace can mean different things to different people and cultures [64–67]. As discussed in a paper by Amadei [19], there is no such thing as a one-size-fits-all unified and optimized static state of peace, the same way as there is no individual united and optimized stationary state of sustainability. It is more realistic to talk about "many [dynamic] peaces" [68] and, along the same line, "many dynamic sustainabilities". What works in a specific context and scale does not necessarily work and translate somewhere else.

This paper focuses on outer peace, which seems to be of higher priority in Western cultures and religions. Another dimension of peace, inner peace at the individual level, is also "an essential component and precondition for a peaceful world". Inner peace is more in line with the world's rich spiritual-religious traditions, such as Hinduism and Buddhism [64]. Regardless of the culture, inner peace is positively correlated with outer peace and is often seen as the place where peace builds outward at the individual and institutional levels [65].

Diamond and McDonald [69] suggested that peace is more than the absence of hostility and violence, and should be considered as "a potential, a possibility, an ever-changing condition [state] . . . a direction in which to head, one step at a time". Peace can be understood as a state or a process. As a state, peace emerges from the interaction of multiple socio-economic, infrastructure, and environmental systems operating in a constrained landscape of specific context and scale. These systems constitute what is referred to in the peace studies and conflict management literature as peace infrastructure [70,71].

In that infrastructure, peace can be defined as "an organizing principle and an enabling violent-free state of dynamic equilibrium emerging from the right relationships among different populations and their interaction with the various systems in the landscape upon which they depend" [19]. This systems-based definition of peace builds on that of sustainability proposed by Ben-Eli [72] as: "an organizing principle and a dynamic [symbiotic] equilibrium in the processes of interaction between a population and the carrying capacity of an environment such that the population develops to express its full potential

without adversely and irreversibly affecting the carrying capacity of the environment upon which it depends". Both aforementioned definitions lend themselves well to using systems modeling tools to capture how the two states of peace and sustainability interact.

As a process, peace unfolds over time through peacebuilding (building conditions for peace), peacemaking (getting parties to find common ground), and/or peacekeeping (supporting sustainable peace) efforts. Another way of looking at peace is to see it as a noun (outcome) or a verb (process), depending on how peace is being addressed. It should be noted that this last remark can also be made about health, sustainability, and resilience.

As noted by Gittins and Velasquez-Castellanos [35], "there are about 35 theories of peace, at least at the university level". The conflict and peace studies literature frequently refers to Johan Galtung's work, who pioneered the concepts of negative and positive peace in the early 1960s [73,74]. In short, negative peace relates to the absence of war and direct or organized violence. Undesirable violence and fear of violence cease to exist due to activities such as "ceasefires, disarmament, prevention of terrorism and state terrorism, nonviolence" [75].

On the other hand, positive peace relates to the presence and prevalence of positive attributes, conditions, and priorities that promote "social and economic justice, environmental integrity, human rights, and development" and contribute to the structural "integration of human society" [73]. As remarked by Fischer [75], positive peace activities may range from "building a life-sustaining economy at the local, national and global level in which everyone's basic needs are met" to "good governance and participation, self-determination, human rights".

Galtung [74] added another dimension of peace (cultural peace) to positive and negative peace. It refers to the "aspects of a culture that serve to justify and legitimize direct [negative] peace and structural [positive] peace". Cultural peace activities may include: "promotion of a culture of peace and mutual learning; global communication and dialogues; development of peaceful deep cultures and deep structures; peace education; peace journalism" [75].

Since peace is not a direct "measurable commodity" [69] and is difficult to conceptualize at different contexts and scales, questions arise as to how to (i) measure it indirectly through indicators and proxies and (ii) monitor and evaluate peace over time. The challenge here is how to measure a state that is the outcome of many interacting systems and sub-systems (e.g., social, economic, environmental, infrastructure) with different levels of complexity, uncertainty, and adaptability, and subject to multiple constraints (geopolitical, environmental, cultural, etc.).

At the national level, a measure of peace was suggested by the Institute for Economics and Peace (IEP) in Sydney, Australia [76–78]. Two indices were proposed to semi-quantify peace: the Positive Peace Index (PPI) and the Global Peace Index (GPI). The IEP considers positive peace as being founded on eight interdependent pillars or domains, each containing three weighted indicators (Figure 1). The domains contributing to positive peace include (i) a well-functioning government; (ii) a sound business environment; (iii) equitable distribution of resources; (iv) acceptance of the rights of others; (v) good relations with neighbors; (vi) a free flow of information; (vii) high levels of human capital; and (viii) low levels of corruption. Mathematically, the PPI is the weighted average of 24 indicators.

A second index called the Global Peace Index (GPI; values of country GPIs can be found at http://visionofhumanity.org/indexes/global-peace-index/, accessed on 10 Janaury 2021) was also proposed by the IEP to measure negative peace (i.e., the level of country peacefulness) for the same countries as the PPI [77]. It consists of 23 indicators (10 external and 13 internal) distributed over three domains: ongoing domestic and international conflict, societal safety and security, and militarization. Analysis of the PPI and GPI overall scores at the country level indicates that both indices are highly positively correlated with a coefficient of correlation of 0.75 for data obtained from 2008–2017 [78].

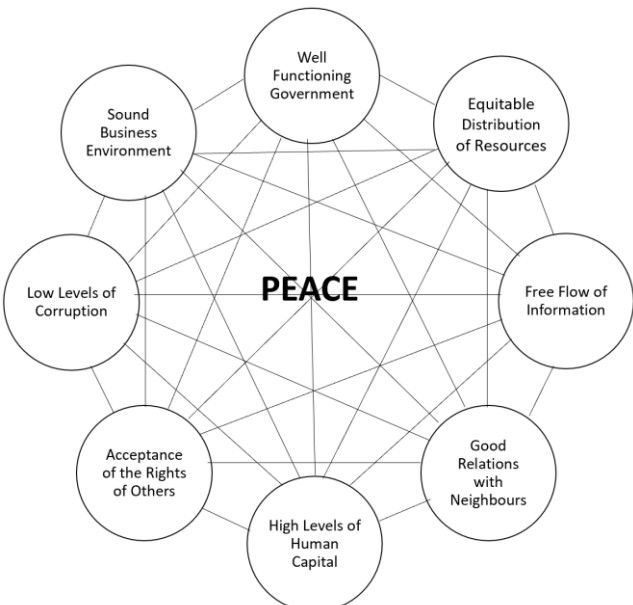

**Figure 1.** The eight pillars (domains) that create positive peace according to the IEP [76–78].

Based on the values of their GPI and PPI ranging between 1 (most positive peace) and 5 (least positive peace), the IEP [78] ranks countries into four potential peacefulness states (i) countries with sustainable peace (high positive and negative peace); (ii) countries with a positive peace deficit (low positive peace and high negative peace) that are likely to experience violence in the future; (ii) countries with a positive peace surplus (high positive peace and low negative peace) with potential to become more peaceful over time; and (iv) countries trapped into violence (low negative and positive peace).

Although not initially proposed by the IEP, a third index, the Cultural Peace Index (CPI), can be introduced to measure cultural peace [79]. Even though its indicators are yet to be determined, the CPI is assumed to range between 1 and 5 for consistency with the PPI and GPI.

The interdependence of the three PPI, GPI, and CPI indices is illustrated in Figure 2 using the peace triangle representation of Galtung [74]. In this diagram, each type of peace influences and depends on the other two. One can interpret the area of that triangle as representing the extent of the enabling environment in which peace unfolds over time. Outside the triangle is the external environment of a specific context that influences the three interacting components of peace over time.

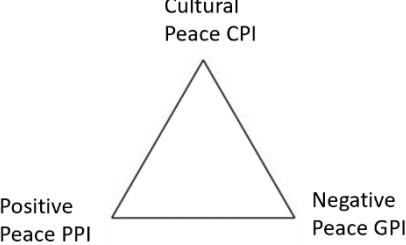

**Figure 2.** The peace triangle. The state of peace unfolds in the triangle (enabling environment), where the three aspects of peace overlap.

Although peace is intimately linked to socio-economic development [80,81], not all forms of peace contribute positively to development and vice-versa. Whether peace is beneficial to development depends mostly on how peace resolution and transformation are designed, implemented, and evaluated. For instance, not addressing the root causes of conflict may result in resuming the conflict and negatively affecting the development in the foreseeable future [82].

Likewise, development can positively or negatively impact peace, depending on the type of development being implemented. It can have intended consequences or create unintended issues [83]. For instance, inappropriate decision-making and trade-offs in activities such as the supply and demand of water, energy, and food resources, inadequate associated infrastructure planning and design, poor decisions in resource management and allocation, and poor governance may result in divisions, unrest, conflict, violence, and insecurity.

In the SDGs sustainable agenda, peace is introduced through SDG 16 (Peace, Justice, and Strong Institutions) and contains 12 targets [5]. Peace also appears in the SDG cross-cutting issues of gender equality, governance, health, inequalities, security, support of vulnerable states, and sustainable cities [84].

## 5. System Dynamics Modeling

### 5.1. Background

Since the 1940s, modeling tools have been proposed in various complexity science disciplines to address ill-defined problems (see the map by Castellani [85]). One of the challenges when modeling complex systems is to select the most appropriate tools to model their dynamics. As noted by Rahmandad and Sterman [86], modeling the dynamics of a given problem depends on "the purpose of the model and the level of aggregation appropriate for that purpose". In short, the selected level of aggregation must match the level of details in the available data sources and provide a balance between "simplicity and realistic depiction of the underlying mechanisms" expected to be at play in the problem of interest. A discussion of the pros and cons of three commonly used modeling methods (system dynamics, discrete event modeling, and agent-based modeling) can be found in Borshchev and Filippov [87].

System dynamics is a branch of systems science originating from Dr. Forrester's work in the 1960s and 1970s [88]. The technique and its multiple applications are well documented in the landmark books by Richmond [89], Sterman [90], and Ford [91], among many others. The value proposition of the SD method in modeling the dynamics of complex systems includes being able to (i) capture both qualitatively and quantitatively how systems continuously change over time due to changes in their components and their interactions; (ii) account for non-linearities, delays, and feedback mechanisms; and (iii) illustrate that as the structure of a system changes, so does its behavior and vice-versa. System dynamics models are causally closed and require selecting closed boundaries. Only endogenous components and factors (those originating from within) are assumed to form the system structure and predominantly dictate the systems' behavior. Compared to other systems modeling tools, the SD method is top-down and can be applied to systems with high aggregation levels (i.e., high abstraction levels). It is appropriate at the strategic level of decision-making [87].

In general, SD models use two types of graphical representations of systems dynamics. Causal loop diagrams (CLDs), not used in this paper, show qualitatively how elements of feedback mechanisms interact causally. The other graphical representation, stock-and-flow diagrams, consists of combining several building blocks (Figure 3) to visualize qualitatively and quantitatively accumulation, flows, delay, and dissipation.

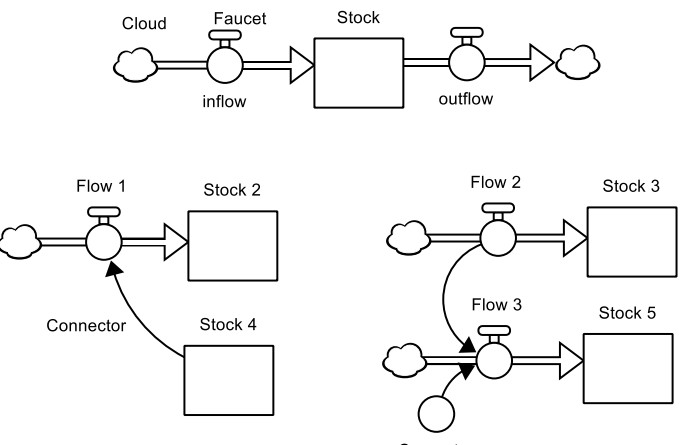

**Figure 3.** Basic building blocks of system dynamics models.

- Stocks correspond to accumulations of something that can be measured at one point in time. They are state variables [88] that define the current state of a system (e.g., peace and development).
- Flow (inflow, outflow) is represented in the form of pipelines (with a faucet controlling the flow). Flow (i.e., flux or rate) results in changes (dynamic behavior) in the stock accumulations and in the entire system. Flows are control variables [88] that create changes in the state (e.g., peace and development) of a system.
- Clouds indicate infinite sources or sinks, somewhere outside of the system boundaries.
- Converters are used to convert or transform information from one stock-and-flow path to another, or to feed information into an existing flow. A converter can also represent a stock if there is no flow in and out of the stock. They are converting variables. Converters can change over time and be described in a functional form.
- Connectors indicate transmission or links of actions and information (i.e., causal connections) between variables such as stock-to-flow, flow-to-flow, or between converters. One or several variables can provide input to and have some influence on another variable through connectors.

In general, system dynamics models (SD) consist of combinations of these five building blocks. However, it must be kept in mind that there is no such thing as a one-size-fits-all SD model that would capture all the possible dynamics in the interaction between peace and development in multiple contexts and scales. We present two examples of SD models to illustrate how to capture the general dynamics at play in the interactions between peace and development. Both were developed using the STELLA Architect software (Version 2.0) by isee systems (www.iseesystems.com, accessed on 15 January 2021). The two models come with two interactive interfaces available online that can be used by the readers to explore different scenarios.

### 5.2. A Simple Peace–Development Nexus Model

Figure 4 shows a simple goal-seeking model of nonlinear interaction between development and peace. The two main dynamic variables (stocks) are the current states of development (D) and peace (P). Both are assumed to involve several endogenous factors that are context-specific and interact within a specified boundary (e.g., country). An interactive user interface for this model can be found on the web (https://exchange.iseesystems.com/public/bernardamadei/peace-development-example-1, accessed on 15 February 2021).

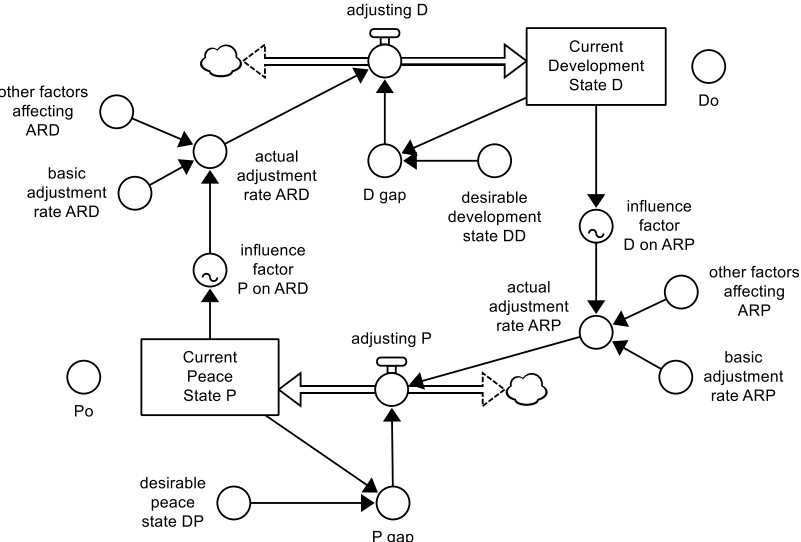

**Figure 4.** Systems dynamics (SD) model of the dynamic between states of development (D) and peace (P).

Starting with an initial baseline (D₀, P₀), both D(t) and P(t) adjust over time toward their respective desirable states of development (DD) and peace (DP) at different adjustment rates, ARD (per year) and ARP (per year), respectively. The actual ARD (P) rate is assumed to be the product of an estimated basic rate and a factor that depends on the current state of peace (~influence factor P on ARD). In Figure 4, the change in the stock D over time ("adjusting D") is equal to the product between ARD(P) and the gap between the desired development state DD and the actual state D as follows:

$$\frac{dD}{dt} = ARD(P) \times (DD - D)$$

Likewise, the actual ARP(D) rate is assumed to be the product of an estimated basic rate and a factor that depends on the state of development (~influence factor D on ARP). In Figure 4, the change in the stock P over time ("adjusting P") is equal to the product between ARP(D) and the gap between the desired peace state DP and the actual state P as follows:

$$\frac{dP}{dt} = ARP(D) \times (DP - P)$$

Note that both ARP(D) and ARD(P) may also depend on other factors (e.g., socio-economic, political, cultural, etc.) that may influence peace and development.

Solving these two nonlinear first-order differential equations with (D₀, P₀) as initial conditions would give an expression for D(t) and P(t) if we knew the functional forms of ARD(P) and ARP(D) and associated parameters and variables (dependent and independent). It is noteworthy that Figure 4 contains two bi-flows instead of two uni-flows feeding the current D and P stocks. The bi-flows model a possible increase or decrease in the two stocks. The current states (D, P) may decrease, for instance, if the 'adjusting D' and 'adjusting P' flow rates become negative. In that case, the development and peace states would degrade over time.

Let us consider that (D, DD), and (P, DP) can be expressed respectively in generic development units (du) and peace units (pu) ranging over two 0–100 scales. Both units are arbitrary and are introduced here as semi-quantitative measures of development and peace. Examples of such measures are discussed in Section 5.4.

The development and peace scales can be broken down into several achievement level groups on an as-needed basis. Each group is specific to the context in which the development–peace nexus analysis is carried out. As an example, the state of development is divided into five levels of development achievement: very low development level (1–20);

low development level (21–40); medium development level with unlikely sustainability (41–60); sustainability possible (61–80); and sustainability likely (81–100). The same approach is used for the state of peace by introducing five levels of peace achievement: very low (1–20), low (21–40), medium (41–60), peace possible (61–80), and peace likely (81–100).

The approach of using a semi-quantitative rating scale to describe the qualitative state of a variable has been used by many authors. For instance, the Institute for Sustainable Infrastructure [92] proposed a framework called *Envision*™ to evaluate and rate the sustainability of infrastructure projects over their life cycle. The rating system consists of five categories with credits. They include (i) quality of life (well-being, mobility, and community); (ii) leadership (collaboration, planning, and the economy); (iii) resource allocation (materials, energy, and water); (iv) the natural world (siting, conservation, and ecology); and (v) climate and risks (emissions and resilience). Points are assigned to each credit for different project sustainability achievement levels: improved, enhanced, superior, conserving, or restorative. Each achievement level has specific characteristics.

Similarly, Schweitzer and Mihelcic [93] proposed an assessment tool to score the sustainability of rural water systems in the developing world. It is based on eight indicators: the activity level, participation, governance, tariff payment, accounting transparency, financial durability, repair service, and system function. Based on the indicators' values, rural water systems are scored into three groups: sustainability likely, sustainability possible, and sustainability unlikely.

Finally, Bouabid and Louis [94] considered eight categories of capacity involved in delivering municipal sanitation services: service level, institutional, human resources, technical resources, economic and finances, energy, environmental, and social and cultural. Each category consists of several requirements. Each requirement is rated with a score ranging between 0 and 100, broken down into five rating groups with 20 units each. For each category of capacity, a capacity factor is calculated as the weighted sum of its requirement scores. The lowest capacity factor is understood as the most vulnerable place in the community where intervention to improve a specific service is first needed. According to Bouabid and Louis [94], it can be interpreted as a semi-quantitative measure of the stage of development of a community and its readiness to provide the service. Based on the value of the lowest capacity factor, the stage of development varies between 1 (no capacity) and 5 (capacity to manage centralized systems).

An SD analysis was carried out assuming that (i) the development and peace levels are initially low with $D_o$ = 20 development units (du) and $P_o$ = 10 peace units (pu); (ii) DD = 100 du and DP = 100 pu; (iii) the basic ARD = 0.02/year, the basic ARP = 0.01/year; (iv) P does not impact ARD; and (v) D does not impact ARP. The other factors affecting ARP and ARD remain constant and equal to 1. Figure 5 shows the corresponding asymptotic increase in D and P toward their desired values. Peace increases from being initially low to reach a low to medium achievement level. Likewise, development starts at a low achievement level and reaches a "sustainability possible" level after 50 years.

As a second numerical example, let us consider the case where the "influence factor of P on ARD" and the "influence factor of D on ARP" have functional forms. They are both assumed to vary linearly between −0.2 and 1 as P and D vary between 0 and 100 peace and development units, respectively. In this example, the initial development and peace levels are very low with $D_o$ = 20 du and $P_o$ = 5 pu. Figure 6 shows the variation of D and P over 50 years. Both development and peace decrease over time to very low values.

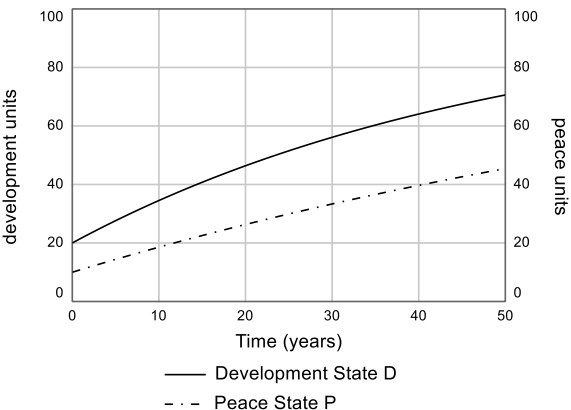

**Figure 5.** A numerical example showing the variation of development and peace states with time. The two influence factors are equal to 1.

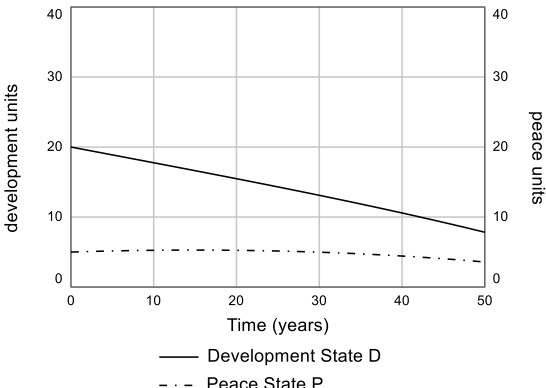

**Figure 6.** A numerical example showing the variation of development and peace states with time. The two influence factors in Figure 3 are linear functions of P and D.

Note that other numerical examples can be conducted with the STELLA-Architect software. The software includes various functionalities that allow users to carry out sensitivity and parametric studies and optimization.

### 5.3. A More Complex Peace–Development Nexus Model

The SD model in Figure 3 represents a simplified picture of the more complex dynamic usually at play between different aspects of the peace and development states at some specific scale and context. The second model shown in Figure 7 generalizes the dynamics of Figure 3 when several interacting development and peace sectors are considered.

To illustrate some possible interactions, we will consider three peace sectors $P_i$ (i = 1–3) (positive, negative, and cultural peace) interacting with three development sectors $D_i$ (i = 1–3) (food security, energy security, and water security). In Figure 6, layered stocks represent the current development and peace states as ($3 \times 1$) arrays.

Compared to Figure 3, the converter "~influence factor P on ARD" is now a $3 \times 3$ array with nine components. Each represents how each peace sector affects the adjustment rate of a development sector in a functional form. Since three peace sectors influence each development sector, a weighted average of that influence is determined in the $3 \times 1$ array converter "weighted influence P on ARD". The user selects the weights. For instance, positive, negative, and cultural peace may affect the adjustment rate in water security differently. The weighted average determines how peace, in general, involves a change in water security. The same can be done for energy and food security. The actual adjustment rate $ARD_i$ (i = 1–3) for each development sector $D_i$ is calculated as the product between

the "weighted influence P on ARD" for that sector and the basic adjustment rate and other factors affecting that sector.

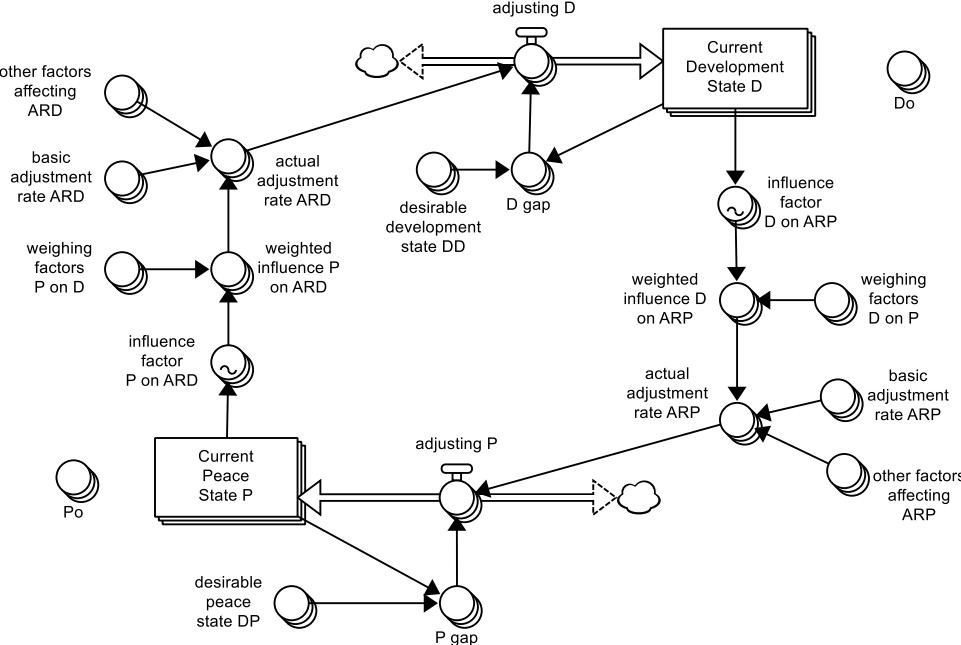

**Figure 7.** SD model of the dynamic between states of development (D) and peace (P). Three sectors of development and peace are considered.

The same approach is used for the "~influence factor D on ARP". The user selects nine functions to describe how each development sector affects the peace sectors. In our case, how water, energy, and food security affect positive, negative, and cultural peace. Since three development sectors influence each peace sector, a weighted average of that influence is determined in the $(3 \times 1)$ array converter "weighted influence D on ARP". For instance, water, energy, and food security may affect the adjustment rates of the three peace sectors differently. The actual adjustment rates $ARP_i$ (i = 1–3) for each peace sector $P_i$ are calculated as the product between the "weighted influence D on ARP" for that sector and the basic adjustment rates and other factors affecting that sector.

A numerical example is shown in Figure 8. In this example, the three peace sectors and the three development sectors' initial values are equal to 5 pu (very low peace) and 20 (low development) du, respectively. All weighing factors P on D and D on P are equal to 1/3. The desirable development and peace sectors values are equal to 100 du and 100 pu, respectively. The basic development and peace adjustment rates are constant and equal to 0.02/year and 0.01/year, respectively. Linear functions were selected to capture the positive influence of peace on development and development on peace. An interactive user interface for this model can be found on the web (https://exchange.iseesystems.com/public/bernardamadei/peace-development-example-2, accessed on 22 February 2021).

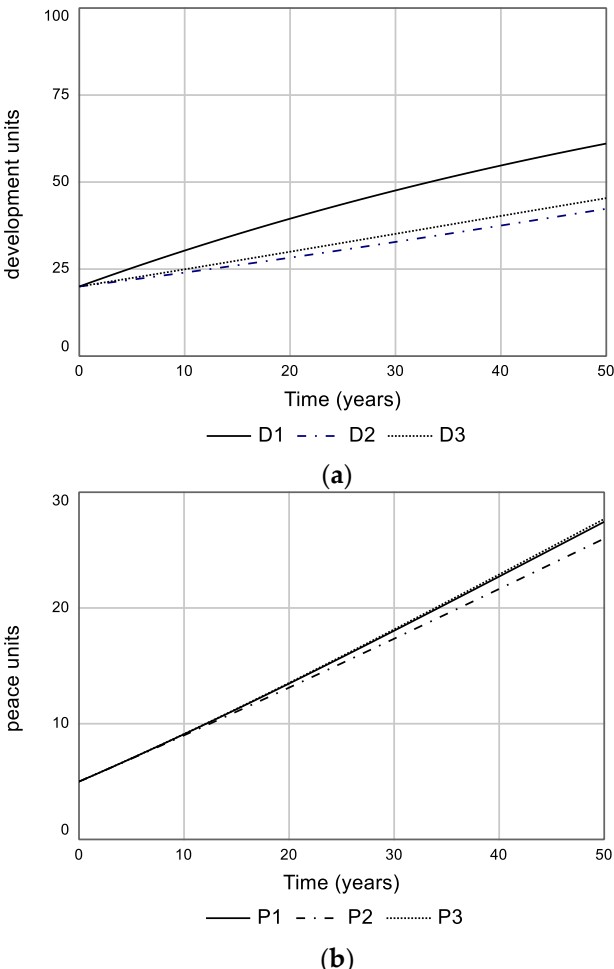

**Figure 8.** Variation of the three sectors of development (**a**) and peace (**b**) with time.

*5.4. Discussion*

The models considered above are two of many possible SD models that could be developed to explore the linkages and feedback mechanisms between development and peace at the country level. Both models are general enough to allow users to consider specific development and peace sectors deemed necessary for the case study.

The arbitrary development units (du) and peace units (pu) used in the two models can be related to existing measures of peace and sustainability. For instance, in the first model, the overall state of peace P could be related to SDG 16, the positive peace index (PPI), the global peace index (GPI), or a combination of PPI and GPI. Likewise, the D variable could be some measure of development. As an example, if food security (SDG 2), energy security (SDG 7), and water resources security (SDG 6) at the country level are assumed to be indicators of development, D could be related to the FEW Security Index proposed by Willis et al. [95]. This index is calculated as the geometric mean of three sub-indices related to the food, energy, and water sectors. Finally, a third option to quantify D is to relate it to the SDG index, which is a recent index introduced to monitor and evaluate in an integrated way the evolution of the 17 SDGs at the country level [40].

In the second model, the development and peace states are represented by multiple sectors. More development sectors can be included in the model, such as various SDGs, the six SDG transformations [7,40], or the five sectors of sustainability [15]. One sector (SDG 16) or multiple sectors (positive, negative, and cultural) can represent the state of peace.

In Figures 3 and 7, the basic adjustment rates of peace and development (ARD and ARP) and the other factors affecting peace and development, can themselves be time-

dependent functions if necessary. Whether they are positive or negative will dictate how one or several development sectors and peace sectors may increase or decrease in value.

It should be noted that the second model does not account for the interaction between the sectors that define the state of development and those that define the state of peace. Accounting for such interactions would require developing more complicated SD models [19].

Finally, it is the experience of the author that one of the challenges of developing system dynamics models of the peace–development nexus is finding a balance between simplicity and a realistic depiction of the structural mechanisms underlying the problem being analyzed. SD models can quickly become overwhelming, as it is easy to fall into selecting too many details that lead to "model paralysis in analysis". Another challenge related to the previous one is finding realistic data sources to estimate the model parameters and their linkages. A third challenge is to recognize that system dynamics modeling is not a random process. As shown in Figure 9, the comprehensive modeling of complex peace–development nexus problems requires the following stages: (i) identification of development and peace problems; (ii) definition of these problems; (iii) formulation of possible SD models using causing loop diagrams and/or stock-and-flow diagrams; (iv) selection of input parameters; (vi) model calibration to reproduce problem current and past dynamics; and (vii) conducting parametric and sensitivity analyses toward (viii) policies and decision making. These eight stages involve multiple feedback loops.

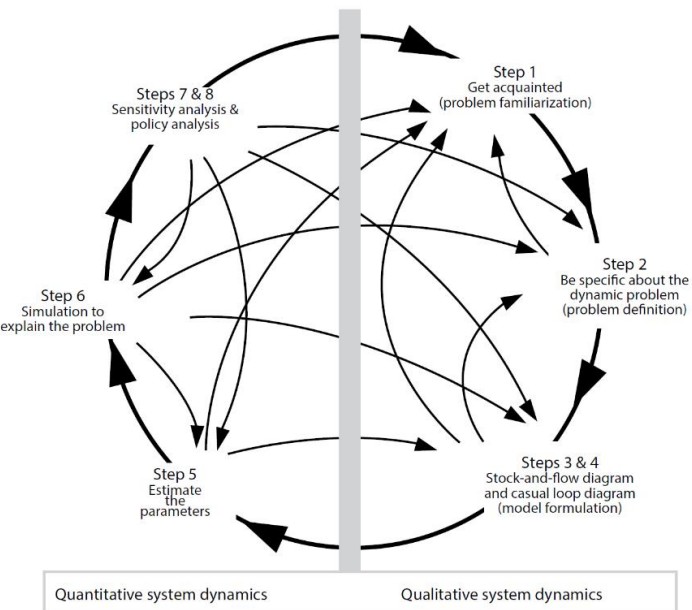

**Figure 9.** Suggested stages in system dynamics modeling (from Ford 2010). The steps consist of those involved in qualitative modeling (right-hand side) and those involved in quantitative modeling (left-hand side).

## 6. Conclusions

Extraordinary times can create unique opportunities. As we enter the third decade of the 21st century, there is a unique opportunity to build back better and develop "dynamic new normals" (if we want to call it that way) that do not bring back the different forms of injustice and inequality (ecological, racial, social, economic, and gender) from the past. Addressing the consequences of the COVID-19 pandemic while dealing with the global challenges facing humanity today and in the near future requires adopting a new mindset. The details of that mindset are yet to be agreed upon and implemented by the international community. As discussed in this paper, some of the characteristics of the mindset include: (i) using an integrated approach to socio-economic development (and humanitarian aid) based on principles of complexity and systems science; (ii) adopting a more mature level

of consciousness in the management and operation of our institutions and occupations; (iii) investing in scientific and technical innovation that embody the five aspects of sustainability (people, planet, profit, partnership, and peace); (iv) developing socio-economic partnerships and collaborations that respect participation and empowerment; and (iv) account for the inner and outer dimensions of human development.

There is a need to reconsider Agenda 2030's priorities for the next ten years and beyond with two goals in mind. An immediate goal is to prevent further decline in socio-economic development that would affect society's poorest sections the most. Another goal is to plan for medium- to long-term sustainability. Meeting both goals requires working on multiple tracks of change simultaneously. As suggested by Moritz [28], they include: repairing what is currently damaged; rethinking change without building back past and current vulnerabilities; reconfiguring development without re-adopting the business-as-usual mindset; and reporting how change takes place through monitoring and evaluation and proposing course correction. To that list, one can add reconnecting with the inner dimension of human development.

Addressing these priorities and developing an action plan that guarantees a certain level of success is not easy. As noted in the TWI2050 [7] report, "success is a matter of choice. Choice requires the deployment of economic, political, and social instruments, technological and cultural innovations, and changes in lifestyles to bring about the needed transformational changes at every scale". Unfortunately, the past 30 years have shown that the lack of will to change from policymakers and practitioners and other geopolitical issues often hinder socio-economic development progress. Inherent to the entire history of socio-economic development are multiple intended and unintended roadblocks that limit progress.

The value proposition of sustainability and peace in the world of tomorrow has become more imperative than ever. Both are entangled. Not effectively addressing SDG 16 may jeopardize all other SDGs [96]. However, addressing how peace interacts with SDGs using a systems approach is not straightforward. As shown in the simple SD models herein, understanding the peace–development nexus requires selecting the sectors that define peace and development. Another challenge is quantifying how development changes the state of peace and how peace changes the state of development. Data and case studies are needed to quantify such complex interactions to be able to consider trade-offs and synergies.

Sustainable development is more than meeting independent goals [97,98]. There is no one-size-fits-all approach to addressing the linkages between the SDGs systematically. All models are context- and scale-specific and are based on an interpretation of reality, but not the reality itself. What works at one scale may not work on another scale.

Finally, a limitation of the SDGs is that they are defined at the country level but cannot be scaled down to other levels. A question remains as to how relevant the SDGs are at the local level when outside experts define them with limited or no input from those who face the actual problems [35].

**Funding:** This research received no external funding.

**Institutional Review Board Statement:** Not applicable.

**Informed Consent Statement:** Not applicable.

**Data Availability Statement:** Not applicable.

**Conflicts of Interest:** The authors declare no conflict of interest.

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
