# Peer review of "Systemic Modeling of the Peace–Development Nexus"

_sustainability, doi:10.3390/su13052522_

Round 1
Reviewer 1 Report
Title : Systemic Modeling of the Peace-Development Nexus
Authors : Bernard Amadei
-----------------------
In this paper, the author presents the strong links between peace and development around the world, and gives a system of non-linear differential equaltions to model these links.
This question is well defined, relevant and seems to be not that original with regard to the bibliography, but the paper is very interesting to read.
Strengths :
. This paper is very well documented (cf. references)
. The subject addressed is very interesting.
. The simulation examples are very illustrative too.
Weakness :
. The model is too rough, and not enough developed in my opinion. The paper would gain in scientific soundness with the presentation and development of these equations.
Significance :
. The content and technical quality of this paper of this paper are correct for me and deserves to be published.
. The conclusion is correctly justified and supported by the results.
. Some limits of the results obtained have been identified, presented and discussed. This is very interesting.
. I took interest and pleasure to read this paper.
Quality of presentation :
. The abstract is clear and presents correctly the subject addressed in this paper.
. The sections of this paper are not conventional for a scientific paper, but the interest of this paper is undoubtful.
. The subheadings used for the redaction of this paper make it clear.
. This paper is clear, easy to follow and to read, and logically written.
. The data and analyses are appropriately presented.
. The conclusion is argumented and clear enough.
Interest to the readers :
. In my opinion, this paper is interesting and deserves to attract a wide readership, beyond the limits of the journal's readership.
Overall evaluation :
. I think there is an overall benefit to publish this work.
. This work provides an advance towards the current knowledge, clearly highlighted in the abstract.
. The authors have addressed an already studied question, with a correct bibliography. More experiments could make a significant gain.
. The English language quality and style of this paper are appropriate and understandable.
As a conclusion, my suggestion to the editor is to accept this paper for publication in Sustainability.
References :
--------------
. 98 research references, out of which 5 self-references, giving a self-reference ratio equal to 5.1%. This is an acceptable ratio.
. The references are cited in the text adequately and appropriately. Particularly, they are described and relevantly presented with regard to the addressed problem.
. The bibliography of this paper is mainly composed of recent references: 20 of them are more than 10 years old, and 78 of them are less than 10 years old.
. Please avoid the formulation 'et al.' in the references section: [41, 44, 46, 47, 51, 55, 58, 61, 95, 96] . The complete list of authors deserve to be cited in this section.
Typos / Comments / Remarks :
------------------------------------
. Line 593: Note 3 at the bottom of the page is empty and does not appear in the text. You can remove it.
. Line 742: 20221 --> 2021.
Author Response
I have improved the presentation of the research ideas. They are now clearly outlined in the abstract.
All Typos / Comments / Remarks were addressed in the revised document.
Reviewer 2 Report
This manuscript lacks some important content. I suggest it is better accepted after some revision. Suggest the authors at least add the following contents:
- Describe the theoretical background of the equation within 476th and 477th lines. I think any reader of this manuscript can't accept that an equation appears suddenly.
- Briefly describe any practical application of models mentioned in this manuscript.
- Within 522th to 524th lines, the authors set D0 = 20 du and P0 = 10 pu. What are the meanings of these du and up?
- The contribution of this manuscript is not clear.
Author Response
- Theoretical background on the fundamentals of system dynamics and the equations was added on pp 10 and 11
- This is submitted to a special issue on systems approach to the SDGs. The paper is about that and a possible methodology to address the goal of the issue.
- du and pu are arbitrary units. See explanation in lines 513-515 and section 5.4. (lines 623-625).
- I have emphasized the contribution of this paper in the revised abstract.
Reviewer 3 Report
This is a timely paper and is quite well written. This paper uses system dynamics modeling to link peace and development. Researchers and industry practitioners could gain insights from this paper when it comes to the challenges of peace & development nexus in different context.
Author Response
I have improved the presentation of the research questions and the SD method. The abstract should now be clearer with presentation of research questions and the why of this paper.